# Motif-Based Graph Representation Learning with Application to Chemical Molecules

Yifei Wang [1,*], Shiyang Chen [2], Guobin Chen [1], Ethan Shurberg [1], Hang Liu [2] and Pengyu Hong [1,*]

1   Department of Computer Science, Brandeis University, Waltham, MA 02453, USA
2   Department of Electrical and Computer Engineering, Stevens Institute of Technology, Hoboken, NJ 07030, USA
*   Correspondence: yifeiwang@brandeis.edu (Y.W.); hongpeng@brandeis.edu (P.H.)

**Abstract:** This work considers the task of representation learning on the attributed relational graph (ARG). Both the nodes and edges in an ARG are associated with attributes/features allowing ARGs to encode rich structural information widely observed in real applications. Existing graph neural networks offer limited ability to capture complex interactions within local structural contexts, which hinders them from taking advantage of the expression power of ARGs. We propose **m**otif **c**onvolution **m**odule (MCM), a new motif-based graph representation learning technique to better utilize local structural information. The ability to handle continuous edge and node features is one of MCM's advantages over existing motif-based models. MCM builds a motif vocabulary in an unsupervised way and deploys a novel motif convolution operation to extract the local structural context of individual nodes, which is then used to learn higher level node representations via multilayer perceptron and/or message passing in graph neural networks. When compared with other graph learning approaches to classifying synthetic graphs, our approach is substantially better at capturing structural context. We also demonstrate the performance and explainability advantages of our approach by applying it to several molecular benchmarks.

**Keywords:** graph neural network; molecular representation; molecular property prediction; graph matching; interpretability; motif-based pretraining; GPU-enabled accelerating

## 1. Introduction

The amount of graph data has grown explosively across disciplines (e.g., chemistry, social science, transportation, etc.), calling for robust learning techniques for modeling knowledge embedded in graphs and performing inference on new graphs. To shed new light on the mechanisms underlying observations, the learning techniques need to be interpretable so that we can link structural patterns to properties of interest. Many types of complex graphs (e.g., chemical molecules, biological molecules, signal transduction networks, multi-agent systems, social networks, knowledge graphs, etc.) can be naturally represented as attributed relational graphs (ARGs) [1,2]. The ARG representation extends ordinary graph representations by associating attributes (or features) with nodes and edges to characterize the corresponding entities and relationships, respectively. This makes ARGs substantially more expressive, which makes them appealing to many real-world applications; however, the nuance of ARGs comes with added complexities in training and analysis. We denote an ARG as $G = < \{v\}, \{e_{uv}\}, \{\mathbf{a}_v\}, \{\mathbf{r}_{u,v}\} >$, where $\{v\}$ is the node set, $\{e_{u,v}\}$ is the relation set with $e_{u,v}$ indicating the relation between nodes $u$ and $v$, and $\mathbf{a}_v$ and $\mathbf{r}_{u,v}$ are the attribute vectors of node $v$ and relationship $e_{u,v}$, respectively.

Recently, graph neural networks (GNNs) [3–6], which operate on the graph domain, have been combined with deep learning (DL) [7] to take advantage of big graph data. Many GNN variants have been proposed for a variety of applications (e.g., visual scene understanding, learning dynamics of physical systems, predicting properties of molecules,

predicting traffic, etc.) [8–18]. In this study, we focus on the application of graph representation learning to efficiently and accurately estimate the properties of chemical molecules, which is in high demand to accelerate the discovery and design of new molecules/materials. In addition, there is an abundance of publicly available data in this domain, for example, the QM9 dataset [19]. In the QM9 dataset, each chemical molecule is represented as an ARG with nodes and relations representing atoms and bonds, respectively. Each node has one attribute storing the atom ID and the 3D coordinates, and each relation has attributes indicating bond type (single/double/triple/aromatic) and length.

Accurate quantum chemical calculation (e.g., typically using density functional theory (DFT)) needs to consider complex interactions among atoms and requires a prohibitively large amount of computational resources, preventing the efficient exploration of vast chemical space. There have been increasing efforts to overcome this bottleneck using GNN variants to approximate DFT simulation, such as, enn-s2s [15], SchNet [20], MGCN [21], DimeNet [22], DimeNet++ [23], and MXMNet [24].

GNNs aim to learn embeddings (or representations) of nodes and relations to capture complex interactions within graphs, which can be used in downstream tasks, such as graph property prediction, graph classification, and so on. The message passing mechanism is widely used by GNNs to approximate complex interactions. A GNN layer updates the embedding of a node $v$ by transforming messages aggregated from its neighbors:

$$\mathbf{a}_v^{(l+1)} = f_1(\mathbf{a}_v^{(l)}, \sum_{u \in \mathcal{N}_v} f_2(\mathbf{a}_u^{(l)}, \mathbf{r}_{uv}^{(l)})) \tag{1}$$

where $l$ indicates the $l$-th GNN layer ($l = 0$ corresponds to the input), $\mathcal{N}_v$ is the neighbor set of node $v$, $\mathbf{a}_v^{(l)}$ is the embedding of node $v$, $\mathbf{r}_{uv}^{(l)}$ is the embedding of relation $e_{uv}$, $f_1$ is the node embedding update function, and $f_2$ is the interaction function passing messages from neighbors. The functions $f_1$ and $f_2$ can be based on neural networks. Relation embedding updates can also be implemented using neural networks to integrate the $l$-th layer embedding of a relation with the $l$- or $(l + 1)$-th layer embeddings of the nodes connected to the relation.

In the context of predicting molecular properties, innovations in GNN variants mainly focus on improving message passing to better utilize structural information. For example, SchNet [20] considers the lengths of relationships (i.e., bonds between atoms) using a band of radial basis functions when calculating message passing. MGCN [21] stacks GNN layers to hierarchically consider quantum interaction at the levels of individual atoms, atom pairs, atom triads, and so on. When calculating the message passing to a target node from one of its neighbors, DimeNet [22] proposes directional embedding to capture interactions between neighboring bond pairs and is invariant in rotation and translation. DimeNet++ [23] improves the efficiency of DimeNet by adjusting the number of embedding layers and the embedding sizes via down-/up-projection layers. MXMNet [24] analyzes the complexity of the directional embedding proposed in DimeNet and decomposes molecule property calculations into local and non-local interactions, which can be modeled by local and global graph layers, respectively. The expensive directional embedding is only used in the local graph layer. In addition, MXMNet proposes efficient message passing methods to approximate interaction up to two-hop neighbors in the global layer and interactions up to two-hop angles in the local graph layer.

Existing GNNs typically start with node attributes, which do not efficiently capture structural information. In addition, each message-passing calculation considers the limited local context of the destination node. Most of the early studies on GNNs treated relations as independent in each iteration of message calculation. DimeNet/DimeNet++ and MXMNet consider the interaction between a one-hop relation and its neighboring two-hop relations. Although MGCN can potentially add higher layers to directly consider larger local contexts, its interaction space will increase exponentially with respect to the layer number. Moreover, it may not be straightforward to choose the number of levels because nodes have different local context sizes. We hypothesize that the local context space can be

well-characterized by a set of motifs, each of which may correspond to a certain type of local structure/substructure. For example, a motif may represent a chemical functional group. The motif set can be learned from data and be used to extract node features that explicitly encode the local context of the corresponding node, and, hence, improve the performance of a GNN. We, therefore, propose a motif-based graph representation learning approach with the following major components: (a) unsupervised pre-training of motifs; (b) motif convolution for isomorphic invariant and local-structure-aware embedding; (c) highly explainable motif-based node embeddings; and (d) a GPU-enabled motif convolution implementation to overcome the high computational complexity. We demonstrate our approach by its application to both synthetic and chemical datasets.

## 2. Motif-Based Graph Representation Learning

The key of our motif-based representation-learning technique is a motif convolution module (MCM) (Figure 1A), which contains a motif convolution layer (MCL) connected to an optional multilayer perceptron (MLP) network. The motifs in an MCL are spatial patterns and can be constructed by clustering subgraphs extracted from training graphs (Figure 1B). These motifs describe various substructures representing different local spatial contexts. The convolution step applies all motifs on every node in an input graph to produce a local-context-aware node representation, which is invariant to transformations (rotation and translation in 3D). The MLP component can further embed the above node representation by exploring interactions between motifs. The node embeddings produced by MCM encode local structural context, which can empower downstream computations to learn richer semantic information. Below, we explain in more details motif vocabulary construction, motif convolution, and using MCM with GNNs.

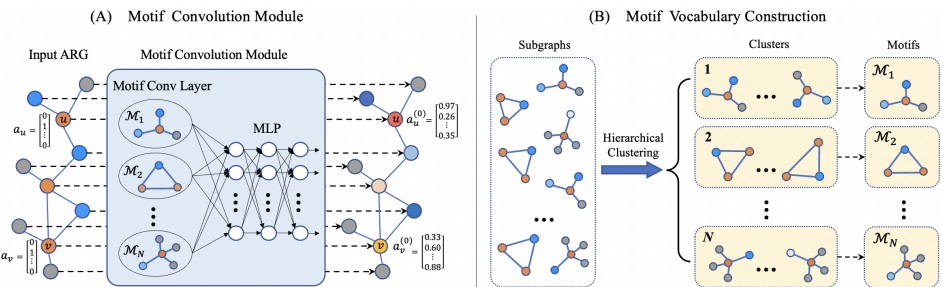

**Figure 1.** Motif convolution module. (**A**) The convolution operation calculates the structural similarity score between every of the $N$ motifs and the subgraph centering at each node in the input graph (see Sections 2.2 and 2.3) to produce an $N$-dimension context-aware representation for the corresponding node, which is further transformed by a multilayer perceptron (MLP) network to produce a MCM-embedding for the input node. For example, although two input nodes, $u$ and $v$, represent the same element (e.g., atom), their MCM-embeddings $a_u^{(0)}$ and $a_v^{(0)}$ are different as $u$ and $v$ are in different local context. An expanded illustration of MCM is shown in Figure 2. The output of MCM can be fed into GNNs. (**B**) The motif vocabulary is built via clustering on subgraphs sampled from input graphs (Section 2.1).

### 2.1. Motif Vocabulary Construction

Ideally, the motif vocabulary should be learned in an end-to-end fashion; however, this would incur an extremely high computational complexity. Therefore, we turned to a straightforward method for building a motif vocabulary that represents recurrent spatial patterns in training ARGs. First, we sampled a large number of subgraphs (e.g., $k$-hop neighborhoods) from the dataset. Each subgraph records its own center node. To make the extracted subgraphs cover local contexts as much as possible, we reduced the probability of sampling a subgraph by 50% if the center node of the subgraph already appears in a sampled subgraph. This allows unvisited local contexts to be sampled with greater probability. Highly similar subgraphs (up to 3D rotation+translation transformations) can be represented by one motif. To achieve this, the sampled subgraphs are grouped

into a user-specified number of clusters using a hierarchical clustering technique using average linkage [25], implemented in the Orange3 library [26]. A representative subgraph is selected from each cluster as a motif. If the size of the whole subgraph set is too big for the hierarchical clustering algorithm, we can randomly partition the whole subgraph set into many smaller subsets, and apply the above procedure to extract representative subgraphs from each subset. The above procedure is then applied to the representative subgraphs extracted from all subsets to obtain the final motifs. Pair-wise similarity calculations are required to perform hierarchical clustering between subgraphs (each of which are ARGs).

### 2.2. ARG Similarity Measurement

We need to measure the similarity between two ARGs when building the motif vocabulary (Section 2.1) and performing motif convolutions (Section 2.3). Such a similarity measurement should be invariant to the permutation of nodes, which requires node-to-node matching between two graphs. In addition, the similarity measurement should not be sensitive to graph sizes. Otherwise, a larger graph could have a higher chance to be more similar to a motif than a smaller graph. Assuming we have the node-to-node matching, which is represented by a matching matrix $\mathbf{M}$, between two ARGs $G_1$ and $G_2$. Each element $\mathbf{M}_{ui} \in \{0, 1\}$ indicates whether node $u$ in $G_1$ matches with node $i$ in $G_2$. Inspired by [27,28], we define the normalized similarity between $G_1$ and $G_2$ as:

$$S(G_1, G_2) = \left( \sum_{u=1}^{n_1} \sum_{i=1}^{n_2} \sum_{v=1}^{n_1} \sum_{j=1}^{n_2} \frac{\mathbf{M}_{ui}\mathbf{M}_{vj}s_1(e_{uv}^{(1)}, e_{ij}^{(2)})}{2\sqrt{l_1 \times l_2}} + \alpha \frac{\sum_{u=1}^{n_1} \sum_{i=1}^{n_2} \mathbf{M}_{ui}s_2(u, i)}{\sqrt{n_1 \times n_2}} \right) \times \frac{1}{1 + \alpha} \quad (2)$$

where $n_1$ and $n_2$ are the numbers of nodes in $G_1$ and $G_2$, respectively. $l_1$ and $l_2$ are the numbers of edges in $G_1$ and $G_2$, respectively, $s_1(e_{uv}^{(1)}, e_{ij}^{(2)})$ is the relation compatibility function measuring the similarity between $e_{uv}^{(1)} \in G_1$ and $e_{ij}^{(2)} \in G_2$, $s_2(u, i)$ is the node compatibility function measuring the similarity between node $u \in G_1$ and node $i \in G_2$. $\alpha$ is the trade-off parameter to balance the contributions from edge similarities and node similarities. Theorem 1 shows that $S(G_1, G_2)$ is independent of graph sizes. A matching matrix $\mathbf{M}$ is required to compute $S(G_1, G_2)$. Finding an optimal matching between two ARGs is an NP problem and has been widely studied. We leave the details of problem definition and the efficient algorithm for finding a sub-optimal $\mathbf{M}$ for Appendix B. We developed a GPU-accelerated matching method with sublinear complexity (see discussions in Appendix B.5).

**Theorem 1.** *If the compatibility functions $s_1(e_{uv}^{(1)}, e_{ij}^{(2)})$ and $s_2(u, i)$ are well-defined and normalized compatibility metrics, $S(G_1, G_2)$ achieves maximum of 1 if and only if $G_1$ and $G_2$ are isomorphic.*

The proof is in Appendix A.

### 2.3. Motif Convolution

The motif convolution layer (MCL) computes the similarity (see Section 2.2) between every motif and the subgraph centered at each node in an input graph. A motif representation of each input node is obtained by concatenating the similarity scores between the subgraph of the node and all motifs. This representation can be fed into a trainable multi-layer perceptron (MLP) with non-linear activation functions (e.g., ReLU) to produce a further embedding that encodes interactions among motif features. We denote this operation as:

$$\mathbf{a}_u^{(0)} = \mathrm{MCM}(u \in G; \{\mathcal{M}_i\}_{i=0}^N)) \quad (3)$$

where $G$ is an input ARG, $u$ is a node in $G$, $\{\mathcal{M}_i\}_{i=0}^N$ represents the motif vocabulary of size $N$, and $\mathbf{a}_u^{(0)}$ is the MCM embedding of $u$. Figure 2 illustrates an expanded illustration of the MCM computation flow. The convolution operation calculates the structural similarity

score between every motif in the motif set $\{\mathcal{M}_i\}_{i=0}^N$ and the subgraph centering at each node in the input graph. For each node in the input ARG, the similarities between all motifs and the local structure of the node are concatenated to produce an $N$-dimension context-aware representation, which encodes the local structural features represented by motifs. The motif feature representation can be further transformed by a trainable multilayer perceptron (MLP) network to produce the final MCM embedding for the input node. If a user chooses to omit the MLP component, the motif feature representation will be the MCM embedding for the input node. Motifs are obtained via a pre-training process described in Section 2.1. The MLP should be trained with the downstream task.

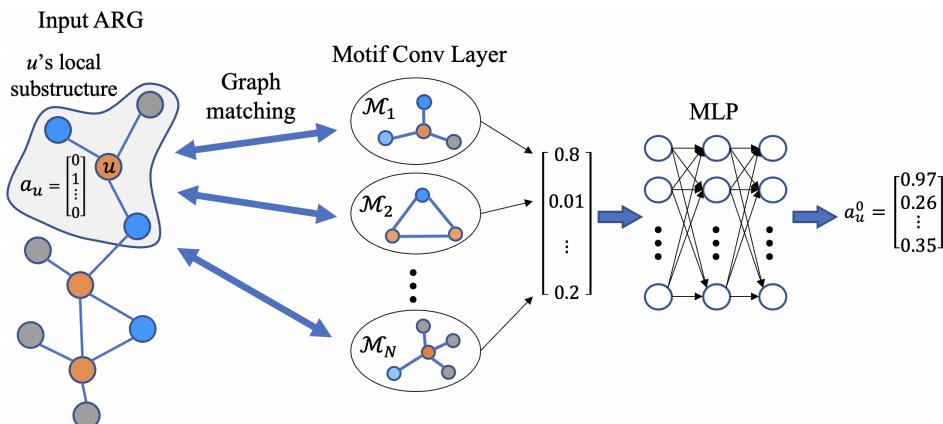

**Figure 2.** Motif convolution module. The convolution operation computes graph matching between each motif and the local structure centering at each node in the input ARG.

### 2.4. Coupling Motif Convolution with GNNs

The MCM can serve as a preceding module for any GNN to form MCM+GNN. The output of the MCM is still an ARG, which can be fed into any GNN that accepts an ARG as input. A readout function of an MCM+GNN model is needed to obtain the final representation of an input $G$:

$$\mathbf{h}_G = \text{READOUT}(\{\mathbf{a}_u^{(L)} | u \in G\}) \tag{4}$$

where $L$ is the number of GNN layers. The READOUT function should be invariant to permutation, and, thus, average, sum, and max-pooling functions are widely used. The final representation $\mathbf{h}_G$ can then be fed into a trainable component (e.g., a fully connected layer or a linear regressor) to generate the desired predictions.

### 3. Experiments and Results

We applied MCM to both synthetic and real data to thoroughly evaluate its potential in classifying graphs, predicting graph properties, and learning semantically explainable representations. All experiments use one-hop neighborhoods in building motifs. The code is available at https://github.com/yifeiwang15/MotifConv (accessed on 4 January 2023).

### 3.1. Classification on the Synthetic Dataset

This experiment shows the advantage of motif convolution in capturing local structural context over GNNs. We designed five ARG templates (Figure 3), and one synthetic dataset of five classes, which share similar node attributes but have different structures. These templates can only be well-distinguished by their overall structures. For example, templates 2 and 5 are very similar to each other except for two edges have different attributes. Sample ARGs were produced from these five ARG templates by randomly adding nodes to templates and adding Gaussian noises of $\mathcal{N}(0, 0.1)$ to node attributes. The number of added nodes in each sample ARG was sampled from a binomial distribution $B(4, 0.1)$. Each sample ARG is labeled by the ID of its ARG template. The task is to predict the

template ID of any given synthetic ARG. We synthesized two datasets of sizes 500 and 10,000, respectively. Each template contributed to 20% of each dataset.

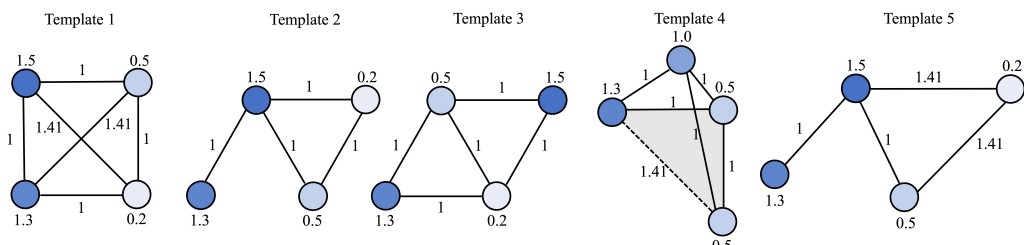

**Figure 3.** Five templates used to generate the synthetic datasets. Template 2 and 5 are designed to make the classification task more challenging, in which only two edges take different attributes.

We only used the MCL of the MCM as it was already sufficient. The readout of the MCL is fed to a logistic regressor (LR) to output the classification result. Standardization was applied to the readout by removing the mean and scaling to unit variance. We named this model MCL-LR. Two readout functions (average pooling and max pooling) were tried, and max pooling always outperformed average pooling. A motif vocabulary of size 5 was constructed. We tried using more than five motifs, and found no significant advantage. We compared MCL-LR with several baseline models built from GNN variants with edge weight normalization implemented by [29], including GCN [30], GIN [31] and GAT [17] (detailed model configurations in Appendix C.2).

We ran each model 20 times on both datasets. In each run, each dataset was randomly split into 8:1:1 for training, validation and test. The average prediction accuracy, as well as the standard deviation, are reported in Table 1. The MCL-LR models significantly outperform other models by an average of 20%. In addition, MCL-LR requires substantially smaller training data as it is able to achieve near-perfect results on the 500 datasets. Furthermore, we observed that the learned motifs illustrated in Figure 4 were quite similar to the underlying templates and contains necessary local structures for discriminant purpose, which explains the superior performance of MCL-LR. The performance by categories (Table A1) suggests that MCL-LR is able to discriminate between highly similar templates, as in the case of templates 2 and 5. In addition, we observed that training of GNNs on the larger dataset took more time and computational resources than MCL-LR.

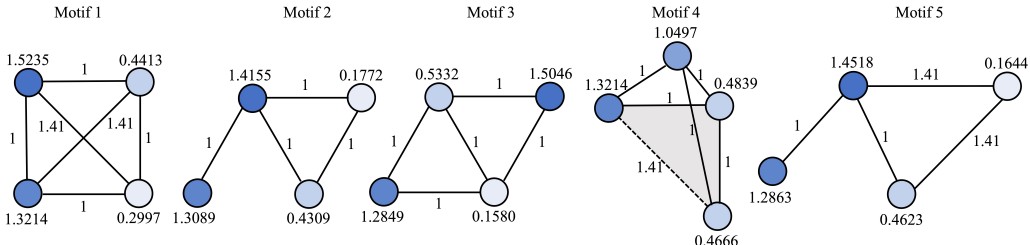

**Figure 4.** The motif vocabulary constructed in the synthetic data experiments. The learned motifs resemble the templates used to generate the synthetic noisy graphs.

**Table 1.** Graph classification results using synthetic data. The best scores are marked as bold.

| Dataset Size | GAT | GCN | GIN | MCL-LR |
|:---:|:---:|:---:|:---:|:---:|
| 500 | $0.691 \pm 0.020$ | $0.745 \pm 0.033$ | $0.640 \pm 0.035$ | $\mathbf{0.996 \pm 0.008}$ |
| 10000 | $0.734 \pm 0.028$ | $0.853 \pm 0.016$ | $0.749 \pm 0.010$ | $\mathbf{0.997 \pm 0.001}$ |

### 3.2. Classification on Molecular Benchmarks

We conducted an experiment using several small- and medium-sized molecular benchmark datasets in MoleculeNet [32]. We compared our model with MICRO-Graph [33] and MGSSL [34] with different generation orders (BFS and DFS), which are also pre-training frameworks for GNNs with a motif-aware fashion. The results demonstrate that MCM can be integrated with GNNs in a broad way. An MCM+GNN model uses the MCM component to preprocess input graphs. We used the open-source package RDKit [35] to parse the SMILES formula of molecules and performed scaffold-split [36,37] to get the train-validation-test split as 8:1:1. Following the suggestions in MGSSL [34], both baseline models (GIN and GCN) have 5-layer with hidden dimension of 300. Mean pooling is used as the readout function after convolutional layers. Both MCM+GCN and MCM+GIN use a motif vocabulary of size 100. Smaller baseline models (3 conv layers and 64 hidden dim in GCN/GIN) are used in MCM-GCN/GIN on all datasets. For each dataset, we carried out five independent runs and reported means and standard deviations. Table 2 shows that GNNs integrated with MCM consistently perform better than the base models. Figure 5 compares the training and test curves of MCM+GIN and GIN, and shows that MCM significantly speeds up and stabilizes training, suggesting MCM+GIN is fundamentally more expressive than GIN. We believe this is because MCM encodes local structural information that is not sufficiently captured with traditional message passing in GNNs. The details of training settings and data preprocessing are provided in Appendix C.4.

**Table 2.** Compare test ROC-AUC (mean ± std) on molecular property prediction benchmarks. The best result for each dataset is in bold.

| Dataset | bace | bbbp | clintox | sider | tox21 | toxcast | hiv |
|---|---|---|---|---|---|---|---|
| GCN | 0.811 ± 0.030 | 0.881 ± 0.036 | 0.615 ± 0.102 | 0.615 ± 0.025 | 0.784 ± 0.017 | 0.633 ± 0.007 | 0.754 ± 0.067 |
| GIN | 0.797 ± 0.049 | 0.873 ± 0.036 | 0.530 ± 0.065 | 0.616 ± 0.025 | 0.783 ± 0.024 | 0.634 ± 0.009 | 0.762 ± 0.058 |
| MICRO-Graph | 0.819 ± 0.004 | 0.870 ± 0.008 | 0.540 ± 0.024 | 0.617 ± 0.018 | 0.774 ± 0.006 | 0.635 ± 0.006 | 0.780 ± 0.026 |
| MGSSL (DFS) | 0.797 ± 0.008 | 0.705 ± 0.011 | 0.797 ± 0.022 | 0.605 ± 0.007 | 0.764 ± 0.004 | 0.638 ± 0.030 | 0.795 ± 0.011 |
| MGSSL (BFS) | 0.791 ± 0.009 | 0.697 ± 0.001 | **0.807 ± 0.021** | 0.618 ± 0.008 | 0.765 ± 0.003 | 0.641 ± 0.070 | 0.788 ± 0.012 |
| MCM + GCN | 0.806 ± 0.026 | **0.917 ± 0.031** | 0.612 ± 0.145 | 0.624 ± 0.024 | 0.794 ± 0.015 | 0.650 ± 0.012 | 0.792 ± 0.046 |
| MCM + GIN | **0.820 ± 0.055** | 0.900 ± 0.031 | 0.655 ± 0.139 | **0.627 ± 0.028** | **0.802 ± 0.015** | **0.651 ± 0.010** | **0.800 ± 0.043** |

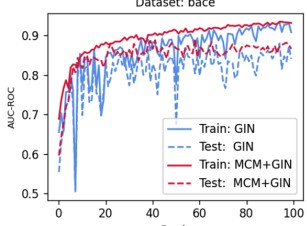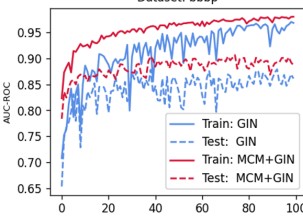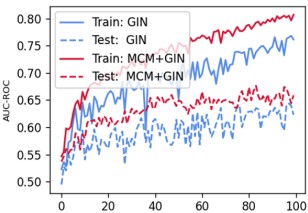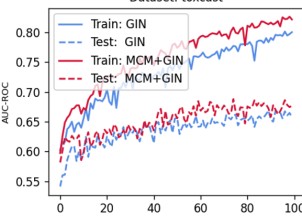

**Figure 5.** The training and testing curves on molecular benchmarks suggest MCM+GIN converge faster and more stably than GIN.

### 3.3. Molecule Property Prediction on QM9

The QM9 dataset [19] is a widely used benchmark for evaluating models that predict quantum molecular properties. It consists of about 130 k organic molecules with up to 9 heavy atoms (C, O, N and F). The mean absolute error (MAE) of target properties is the commonly used evaluation metric. We adopted the data-splitting setting used in [22–24]. More specifically, following [38], we removed about 3 k molecules that failed the geometric consistency check or were hard to converge. We applied random splitting to the dataset, which takes 110,000 molecules for training, 10,000 for validation, and the rest for test. We only used the atomization energy for $U_0$, $U$, $H$ and $G$, by subtracting the atomic reference energies as in [22]. For property $\Delta\epsilon$, we followed the DFT calculation and calculate it by simply taking $\epsilon_{LUMO} - \epsilon_{HOMO}$.

We designed the MCM to be MCL + 2-layer MLP (MLP: input $\rightarrow$ 128 $\rightarrow$ ReLU $\rightarrow$ 128 $\rightarrow$ output). The motif vocabulary size is represented as a hyper-parameter, where we tried 100 and 600 in the experiments. We formed our model MCM+MXMNet by connecting the above MCM to an MXMNet. Two options (5Å and 10Å) were tested for the distance cut-off hyper-parameter $d_g$ of MXMNet. A separate model was trained for each target property and used grid search on learning rate, batch size, motif number, and cut-off distance $d_g$. Edges in molecules are defined by connecting atoms that lie within the cut-off distance $d_g$. Following [22], we did not include auxiliary features like electronegativity of atoms. Detailed training settings are provided in Appendix C.5, and the discussion of motif vocabulary construction and efficiency is in Appendix C.6.

We compared our model MCM+MXMNet with several other state-of-the-art models including SchNet [20], DimeNet [22], DimeNet++ [23] and MXMNet [24]. For other models, we use the results reported in their original works. All experiments were run on one NVIDIA Tesla V100 GPU (32 GB). Table 3 summarizes the comparison results, and shows that our model MCM+MXMNet outperforms others on eight molecule property prediction tasks. For two MXMNet settings, a larger cut-off distance (i.e., $d_g = 10$Å) can lead to better results for some tasks, but not all of them. This is because larger $d_g$ leads to a larger receptive field and thus helps to capture longer range interactions. However, higher $d_g$ might cause redundancy or oversmoothing in message passing and will also increase computation cost. We observed a similar phenomenon for MCM+MXMNet. We also observed that under the same $d_g$ setting, MCM+MXMNet tends to perform better than MXMNet. We believe that this is because MCM helps to produce more informative node representations that better encode local chemical context.

**Table 3.** Comparison of MAEs of targets on QM9 dataset for different tasks. The best result for each task is in bold.

| Task | SchNet | DimeNet | DimeNet++ | MXMNet $d_g = 5$Å | MXMNet $d_g = 10$Å | MCM+MXMNet $d_g = 5$Å | MCM+MXMNet $d_g = 10$Å |
|---|---|---|---|---|---|---|---|
| $\mu$ (D) | 0.033 | 0.0286 | 0.0297 | 0.0382 | 0.0255 | 0.0375 | **0.0251** |
| $\alpha(a_0^3)$ | 0.235 | 0.0469 | **0.0435** | 0.0482 | 0.0465 | 0.0477 | 0.0456 |
| $\epsilon_{HOMO}$ (meV) | 41 | 27.8 | 24.6 | 23.0 | 22.8 | **21.9** | 22.6 |
| $\epsilon_{LUMO}$ (meV) | 34 | 19.7 | 19.5 | 19.5 | 18.9 | **18.5** | 18.6 |
| $\Delta\epsilon$ (meV) | 63 | 34.8 | 32.6 | 31.2 | **30.6** | 32.1 | 31.9 |
| $\langle R^2 \rangle(a_0^2)$ | **0.073** | 0.331 | 0.331 | 0.506 | 0.088 | 0.489 | 0.124 |
| ZPVE (meV) | 1.7 | 1.29 | 1.21 | 1.16 | 1.19 | **1.14** | 1.18 |
| $U_0$ (meV) | 14 | 8.02 | 6.32 | 6.10 | 6.59 | **5.97** | 6.49 |
| $U$ (meV) | 19 | 7.89 | 6.28 | 6.09 | 6.64 | **6.02** | 6.51 |
| $H$ (meV) | 14 | 8.11 | 6.53 | 6.21 | 6.67 | **6.01** | 6.50 |
| $G$ (meV) | 14 | 8.98 | 7.56 | 7.30 | 7.81 | **7.13** | 7.54 |
| $c_v(\frac{cal}{molK})$ | 0.033 | 0.0249 | 0.0230 | **0.0228** | 0.0233 | 0.0230 | 0.0234 |

### 3.4. Explainability of Motif Convolution

The embeddings that MCM learns are highly explainable and encode domain semantics. We visualize the representations of carbons produced by an MCM with 600 motifs in the QM9 experiment. The visualization is created using the T-distributed Stochastic Neighbor Embedding (t-SNE) algorithm [39]. We randomly sampled 15,000 molecules from the QM9 dataset, and then randomly selected 2 carbons from each chosen molecule. Figure 6 shows the t-SNE visualization of these 30,000 atoms' representations learned by MCM. To better understand our representation, we manually labelled 300 carbons randomly sampled from the above 30,000 carbons according to their one-hop local structures. We observed that carbons in the same local context tend to cluster together and are separated from those in different local structures.

More interestingly, we observe that node representations learned by MCM encode meaningful chemical properties. For example, the carbons (red in Figure 6A) in the Trifluoromethyl ($-CF_3$) groups are tightly clustered together, actually stacked into one point. It is known that the more fluorines are connected to a carbon, the shorter the bonds from

this carbon [40], which makes the Trifluoromethyl groups very different from other sub-structures. Moreover, Methylene (-CH₂-) is the most common 'bridge' in organic chemistry, connecting all kinds of functional groups (R, R'). Hence, the carbons (pink in Figure 6A) in the Methylene groups are scattered apart because of their diverse contexts. The carbons in the alcohol functional groups (-CH₂OH, green in Figure 6A) are clustered into two separate sub-groups. This is because they are connected to two very different chemical structures (Figure 6B): cyclic functional groups and linear functional groups.

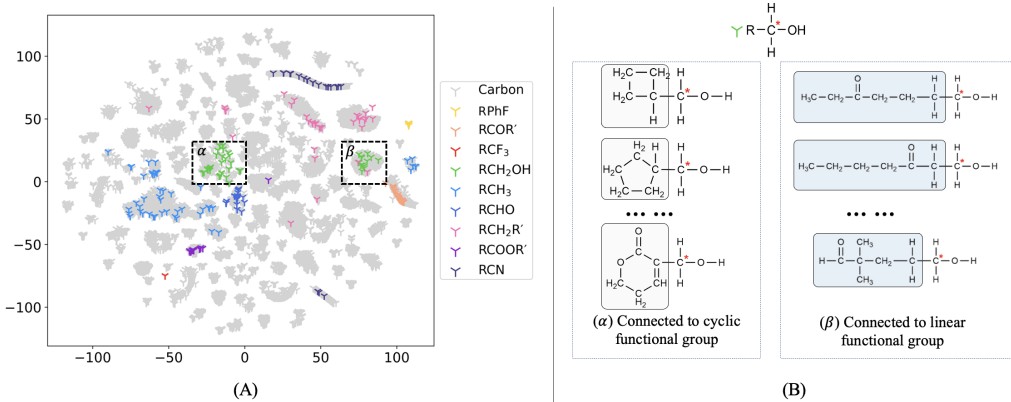

**Figure 6.** Node embeddings learned by MCM. (**A**): The t-SNE visualization of carbon representations learned by MCM. There are 30,000 carbons randomly sampled from the QM9 dataset. Among them, 300 are randomly chosen and are colored based on types of functional groups that carbons belong to, for example, alcohol(-OH) in green, three fluorines ($F_3$) in red, and so on. Both R and R' are the abbreviations for groups in the rest of a molecule. The details of the nine local structure groups are listed in Table A4. The green group are separated into two sub-groups ($\alpha$ and $\beta$). (**B**): The carbons, whose representations visualized in the Left, are marked by red *. The carbons in the green group share the same one-hop local structures shown at the top. The two green sub-groups have distinct characteristics in their at-large local structures. In the $\alpha$ cluster, the marked carbons are connected to cyclic functional groups. In the $\beta$ cluster, the marked carbons are connected to linear functional groups.

## 3.5. Efficiency of GPU Accelerated Motif Convolution

The highest workload in MCM comes from matching motifs with subgraphs, which can be sped up tremendously using parallel computing in GPUs. We developed a CUDA-enabled graph-matching kernel (Appendix B.4) for matching multiple Motif-ARG pairs concurrently, which offer an essential boost to this work. We tested the efficiency of our graph matching kernel under various settings. All experiments were run on NVIDIA GeForce RTX 2080 11GB GPUs. We created four test datasets with graph sizes of 10, 15, 20, and 25, respectively. Each set contains 500 molecules sampled from the QM9 dataset. We ran our CUDA-enabled graph matching kernel using up to eight GPUs to compute pair-wise matching within each dataset. In total, there are 124,750 pairs. The execution times (including loading data from hard disks) of different settings are compared in Figure 7. In general, as expected, it took longer to match larger ARGs. More GPUs help to accelerate the computation. When using a number of GPUs ≤ 4, doubling GPU devices approximately reduced the execution time by half, which indicates that our kernel achieved a balanced workload in parallel. Using more than 5 GPUs only offered marginal speed improvements because GPUs spent significant amounts of time waiting for data to be loaded.

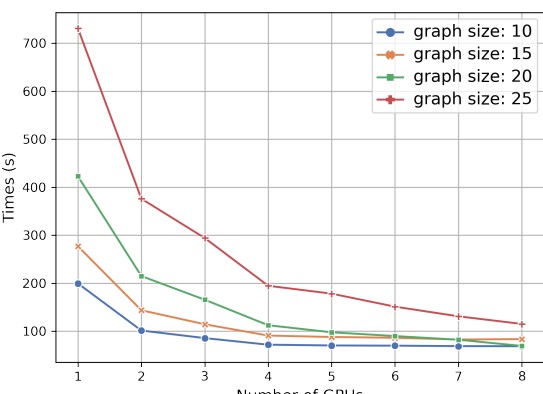

**Figure 7.** Test speed of pair-wise matching on GPUs. Each dataset contains 500 molecular graphs.

## 4. Related Works

Early graph embedding methods [41–43] preserve local neighborhoods of nodes by using biased random-walk based objectives. Some other works, such as [44–46], train node encoders by maximizing the mutual information between local and global representations. These methods encourage the preservation of vertex proximity (i.e., nearby nodes to have similar embeddings) and were originally designed and evaluated for node- and edge-level predictions. However, such methods do not work well for predicting graph-level properties (e.g., molecular properties) since they over-emphasize vertex proximity at the expense of global structural information. For instance, random-walk based methods [41–43] consider limited substructures (e.g., subtrees) as graph representatives. There are several other efforts [47–49] for capturing the structural identity of nodes. However, the applications of such approaches are limited because of their rigid notions of structural equivalence.

Recently, self-supervised approaches were proposed for pre-training GNNs [33,36,50–59]. Self-supervised tasks at node, edge and graph levels were carefully designed to learn general structural and semantic representations that can be fine-tuned for downstream tasks. These approaches broadly fall into two categories. The first one trains models to predict randomly masked-out node attributes [36] or subgraphs [50]. The second one adopts contrastive learning to maximize representation consistency under perturbations [33,56,57,59]. However, these approaches cannot capture the rich information in subgraphs or graph motifs. A few works have been reported to leverage motif-level information. For example, early works such as [47,48] encode local structures as binary properties, which do not reflect deformations of local structures that can happen naturally. Domain knowledge is used to extract motifs and treat them as identifiers [52]. MICRO-Graph [33] is a motif-driven contrastive learning approach for pretraining GNNs in a self-supervised manner. MGSSL [34] incorporates motif generation into self-supervised pre-learning for GNNs. There is much room for improvements to take advantages of local structural information and produce highly explainable node representations. The challenge in motif-based approaches mainly comes from the difficulty in efficiently measuring similarities between input graphs and the automatic construction of a high quality motif vocabulary.

## 5. Discussion

The main contribution of this study is the design of the motif convolution module (MCM). MCM first takes motif discovery from a dataset and applies motif convolution to extract initial context-aware representations for the nodes in input ARGs, which are then embedded in higher level representations using neural network learning. To leverage the power of existing GNNs and target particular applications (e.g., graph classification or regression applications), MCM can be connected as a preceding component to any GNN. One key computational step in MCM is matching ARGs, which is NP-hard in theory and

has sub-optimal solutions. To make it possible to apply MCM to large-scale graph datasets, we modified a graduated assignment algorithm for matching ARGs and implemented a CUDA-enabled version. Currently, the motifs in MCM are fixed once constructed. In our future work, we will develop motifs that are co-trainable with the rest of a model.

## 6. Conclusions

This work presents MCM, a novel motif-based representation learning technique that can better utilize local structural information to learn highly explainable representations of ARG data. To our best knowledge, this is the first motif-based learning framework targeting graphs that contain both node attributes and edge attributes. We show that our approach achieves better results than the state-of-the-art models in a graph classification task and a challenging large-scale quantum chemical property prediction task. Moreover, experimental results highlight the ability of MCM to learn context-aware explainable representations. Motif convolution offers a new avenue for developing new motif-based graph representation learning techniques.

**Author Contributions:** Conceptualization, Y.W., S.C., H.L. and P.H.; methodology, Y.W., S.C., H.L. and P.H.; software, Y.W. and S.C.; validation, Y.W., S.C., G.C., H.L. and P.H.; formal analysis, Y.W.; investigation, Y.W.; resources, H.L. and P.H.; writing—original draft preparation, Y.W., S.C., G.C., E.S., H.L. and P.H.; writing—review and editing, Y.W., S.C., G.C., E.S., H.L. and P.H.; visualization, Y.W., S.C., G.C. and E.S.; supervision, H.L. and P.H.; project administration, H.L. and P.H.; funding acquisition, P.H. All authors have read and agreed to the published version of the manuscript.

**Funding:** This research was funded by NSF DMR 1933525 and NSF OAC 1920147.

**Institutional Review Board Statement:** Not applicable.

**Informed Consent Statement:** Not applicable.

**Data Availability Statement:** We made the synthetic dataset available on https://github.com/yifeiwang15/MotifConv/tree/main/MCM_for_syn, (accessed on 4 January 2023). Datasets from MoleculeNet and QM9 are open source.

**Conflicts of Interest:** The authors declare no conflict of interest.

## Abbreviations

The following abbreviations are used in this manuscript:

| | |
|---|---|
| ARG | Attributed relational graph. |
| GNN | Graph neural network. |
| MLP | Multiple layer perceptron. |
| MCL | Motif convolution layer. |
| MCM | Motif convolution module. |

## Appendix A. Proof of Theorem 1

**Proof.** First let us give a formal definition of well-defined and normalized compatibility metric $s(x_1, x_2) \in [0, 1]$ in the theorem, where $x_1$ or $x_2$ are vectors of the same dimension. It takes a maximal value of 1 if and only if $x_1 = x_2$. One example could be $s(x_1, x_2) = \exp(-\frac{||x_1 - x_2||^2}{2})$.

**Necessity**. The first proof is that if $G_1$ and $G_2$ are isomorphic, $S(G_1, G_2)$ achieves maximum of 1. Obviously, $G_1$ and $G_2$ have the same number of nodes and edges given the isomorphism condition ($n_1 = n_2$ and $l_1 = l_2$). Without loss of generality, we could assume the node ordering in two graphs are the same and the matching matrix $\mathbf{M}$ is the identical

matrix $\mathbf{I}$. Otherwise, we could find a permutation matrix $\mathbf{P}$ to reorder nodes such that $\mathbf{PM} = \mathbf{I}$. Then let us look at the two parts in computing $S(G_1, G_2)$ from Equation (2)

$$\alpha \frac{\sum_{u=1}^{n_1} \sum_{i=1}^{n_2} \mathbf{M}_{ui} s_2(u, i)}{\sqrt{n_1 \times n_2}} = \frac{\alpha}{n_1} \sum_{i=1}^{n_1} s_2(i, i) = \alpha \tag{A1}$$

$$\sum_{u=1}^{n_1} \sum_{i=1}^{n_2} \sum_{v=1}^{n_1} \sum_{j=1}^{n_2} \frac{\mathbf{M}_{ui} \mathbf{M}_{vj} s_1(e_{uv}^{(1)}, e_{ij}^{(2)})}{2\sqrt{l_1 \times l_2}} = \sum_{i=1}^{n_1} \sum_{j=1}^{n_1} \frac{s_1(e_{ij}^{(1)}, e_{ij}^{(2)})}{l_1} = 1 \tag{A2}$$

The last equation holds because the number of edges is $l_1$ and $s_1(e_{ij}^{(1)}, e_{ij}^{(1)})$ takes 1 if edge $e_{ij}$ exists, otherwise 0.

Thus, $S(G_1, G_2) = \frac{1+\alpha}{1+\alpha} = 1$ and we finish this proof.

**Sufficiency**. Another proof is that suppose $S(G_1, G_2) = 1$, then $G_1$ and $G_2$ are isomorphic. We will prove by contradiction.

First let us prove that $S(G_1, G_2) < 1$ if $n_1 \neq n_2$ or $l_1 \neq l_2$. (We assume $n_1 \geq n_2$ without loss of generality.)

Since $\mathbf{M}$ is the hard matching matrix, there is at most one nonzero element (taking value 1) per row and per column, defining an injective function $\phi_\mathbf{M}$ such that $\phi_\mathbf{M}(i) = u$ if $\mathbf{M}_{ui} = 1$. Thus, we have

$$\alpha \frac{\sum_{u=1}^{n_1} \sum_{i=1}^{n_2} \mathbf{M}_{ui} s_2(u, i)}{\sqrt{n_1 \times n_2}} = \alpha \frac{\sum_{i=1}^{n_2} s_2(\phi_\mathbf{M}(i), i)}{\sqrt{n_1 \times n_2}} \leq \alpha \frac{n_2}{\sqrt{n_1 \times n_2}} \leq \alpha. \tag{A3}$$

and

$$\sum_{u=1}^{n_1} \sum_{i=1}^{n_2} \sum_{v=1}^{n_1} \sum_{j=1}^{n_2} \frac{\mathbf{M}_{ui} \mathbf{M}_{vj} s_1(e_{uv}^{(1)}, e_{ij}^{(2)})}{2\sqrt{l_1 \times l_2}} = \frac{\sum_{i=1}^{n_2} \sum_{j=1}^{n_2} s_1(e_{\phi_\mathbf{M}(i)\phi_\mathbf{M}(j)}^{(1)}, e_{ij}^{(2)})}{2\sqrt{l_1 \times l_2}} \leq \frac{2\min(l_1, l_2)}{2\sqrt{l_1 \times l_2}} \leq 1. \tag{A4}$$

where the first inequality holds because $s_1(e_{\phi_\mathbf{M}(i)\phi_\mathbf{M}(j)}^{(1)}, e_{ij}^{(2)})$ takes maximum of 1 given edge $e_{\phi_\mathbf{M}(i)\phi_\mathbf{M}(j)}^{(1)}$ in $G_1$ is identical to edge $e_{ij}^{(2)}$ in $G_2$ and takes 0 if either edge not exists; thus, there are, at most, $2\min(l_1, l_2)$ nonzero terms in the summation.

Strict inequality in the last line of Equation (A3) holds if $n_1 \neq n_2$ and strict inequality in the last line of Equation (A4) holds if $l_1 \neq l_2$. Thus, $S(G_1, G_2) < \frac{1+\alpha}{1+\alpha} = 1$ if either $n_1 \neq n_2$ or $l_1 \neq l_2$. Thus, we complete the first proof.

Next let us prove $G_1$ and $G_2$ are isomorphic by contradiction. Note that we already have $n_1 = n_2$ and $l_1 = l_2$. Without loss of generality, let us assume the matching matrix $\mathbf{M}$ is the identical matrix $\mathbf{I}$; otherwise, we could introduce a permutation matrix to reorder nodes. Then, the injective function $\phi_\mathbf{M}(i) = i$ becomes identical mapping.

If $G_1$ and $G_2$ are not isomorphic, at least one of the following cases must hold:

Case1. ($i$-th node in $G_1$ is not identical to $i$-th node in $G_2$) $\exists i \in [1, 2, \ldots, n_2]$, such that $s_2(i, i) < 1$. Thus, Equation (A3) takes strict inequality.

Case2. (Edge $e_{ij}^{(1)}$ in $G_1$ is not identical to edge $e_{ij}^{(2)}$ in $G_2$, or either edge does not exist.) $\exists i, j$, such that $s_1(e_{ij}^{(1)}, e_{ij}^{(2)}) < 1$. Thus, Equation (A4) takes strict inequality.

For either case, we obtain the strict inequality and, thus, $S(G_1, G_2) < \frac{1+\alpha}{1+\alpha} = 1$, which leads to contradiction. $\square$

## Appendix B. GPU-Enabled ARG Matching

### Appendix B.1. ARG Matching Used in MCM

The convolution operation calculates the structural similarity score between the motif $M_i$ from Motif Conv Layer and $u$'s local substructure from the input ARG. Before taking convolution, we should find the optimal matching assignment between $M_i$ and $u$'s local

subgraph. A graph matching problem is NP-hard and has been well-studied for a couple of years. In the following section, we briefly introduce the problem definition and efficient approximated solutions proposed by [27,28]. To make the computation of graph matching more efficient in practice and meet the need of high-frequency calculation in MCM, we proposed a CUDA-enabled method to accelerate ARG matching, which could achieve 10,000× the speed running on CPUs.

*Appendix B.2. ARG Matching*

It should be noted that finding the optimal matching between two ARGs is NP-hard and can be formulated as a quadratic assignment problem (QAP) [60]. Basically, the optimal matching can be found by solving the following optimization problem:

$$\max_{\mathbf{M} \in \mathbf{R}^{n_1 \times n_2}} \quad \frac{1}{2} \sum_{u=1}^{n_1} \sum_{i=1}^{n_2} \sum_{v=1}^{n_1} \sum_{j=1}^{n_2} s_1(e_{uv}^{(1)}, e_{ij}^{(2)}) \mathbf{M}_{ui} \mathbf{M}_{vj} + \alpha \sum_{u=1}^{n_1} \sum_{i=1}^{n_2} s_2(u, i) \mathbf{M}_{ui},$$

$$\text{s.t.} \quad \forall u \sum_{i=1}^{n_2} \mathbf{M}_{ui} \leq 1, \forall i \sum_{u=1}^{n_1} \mathbf{M}_{ui} \leq 1, \forall u, i \; \mathbf{M}_{ui} \in \{0, 1\} \tag{A5}$$

where $s_1(e_{uv}, e_{ij})$, $s_2(u, i)$, and $\alpha$ are the same to the ones in Equation (2) in the main body. A graduated-assignment-based algorithm was proposed in [27] for finding a sub-optimal matching solutions between two ARGs. A simplified verion of this algorithm was proposed in [28], which runs much faster with little compromise in accuracy. Nevertheless, the matching matrix solved by [28] does not always fulfill the constraints in Equation (A5) in the main body, and may produce ambiguous matching results. We develop a greedy iterative method that converts the soft matching matrix **M** into a hard matching matrix (i.e., containing binary values). Our method finds the maximum in **M**, sets it to 1, and sets all other elements in the same row and column to 0. This step is applied to the rest of **M** until the sum of every row/column in **M** is at most 1.

The above graph matching algorithm still incurs a substantial computational cost when applied to large-scale graph datasets (e.g., the QM9 dataset). We, therefore, implemented a version accelerated by GPU computing, which makes it possible for us to apply MCM to large-scale datasets. The efficiency of our GPU-enabled ARG matching algorithm was discussed in Section 3.5.

*Appendix B.3. Simplified Graduated Assignment Algorithm for ARG Matching*

The graduated assignment algorithm [27] finds sub-optimal graph matching solutions by iteratively solving the first-order Taylor expansion of QAP (Equation (A5)). A simplified graduated assignment algorithm was later proposed by [28] (pseudo codes included in Algorithm A1). It first finds the soft assignment matrix that relaxes the constraint $\mathbf{M}_{ai} \in \{0, 1\}$ to lie in the continuous range $[0, 1]$, then converts it into hard assignment in a greedy way. Algorithm A1 shows the iteration steps to obtain the approximated assignment matrix. Given the initialization of $\mathbf{M}^0$, the objective function $E(\mathbf{M})$ in Equation (A5) can be approximated via a Taylor expansion at $\mathbf{M}^0$; thus, the original optimization problem is equivalent to the assignment problem that maximizes $\sum_{a=1}^{n_1} \sum_{i=1}^{n_2} \mathbf{Q}_{ai} \mathbf{M}_{ai}$, where $\mathbf{Q}_{ai} = \left. \frac{\partial E(\mathbf{M})}{\partial \mathbf{M}_{ai}} \right|_{\mathbf{M}=\mathbf{M}^0}$ is the partial derivative. The optimal **M** at the current step will substitute back as the new initialization and repeat the Taylor approximation period until convergence.

One efficient way to solve an assignment with a constraint (row or column summed up to 1) is by taking softmax with control parameter $\beta > 0$ along with the constrained rows/columns, so that $\mathbf{M} = \text{softmax}(\beta \mathbf{Q})$. Increasing $\beta$ will push the elements in **M** to be either 0 or 1 and result in a hard matching when $\beta \longrightarrow \infty$. However, the assignment problem in ARG matching has two constraints (both row and column summed up to 1). To achieve them, the solver can first perform the element-wise exponential operation such that $\mathbf{M}_{ai} = \exp(\beta \mathbf{Q}_{ai})$, and then alternatively normalize the rows and columns until

convergance to a doubly stochastic matrix (i.e., a soft assignment between two input ARGs) [61]. We initialize $\beta$ with $\beta_0$, and increase it at a rate $\beta_r$ at each iteration until *beta* reaches a threshold $\beta_f$. Finally, the soft assignment result $\mathbf{M}$ is converted into a hard assignment by the greedy procedure explained in Appendix B.2.

---

**Algorithm A1** Simplified graduated assignment for ARG matching.

1: **Input:** $G_1, G_2, \beta_0, \beta_r, \beta_f$
2: **Output:** Hard assignment matrix $\mathbf{M}^*$
3: $\beta = \beta_0$                                                       ▷ Initialize $\beta$.
4: $\mathbf{M}_{ui} = s_1(u,i), \forall u \in G_1, \forall i \in G_2$                         ▷ Initialize $\mathbf{M}$.
5: **while** $\beta \leq \beta_f$ **do**
6:     $\forall u \in G_1, \forall i \in G_2$
7:     $\mathbf{Q}_{ui} = \frac{1}{2} \sum_{v=1}^{n_1} \sum_{j=1}^{n_2} s_2(e_{uv}, e_{ij}) \mathbf{M}_{vj} + \alpha s_1(u,i)$      ▷ Taking the partial derivative.
8:     $\mathbf{M}_{ui} = \exp(\beta \mathbf{Q}_{ui})$                        ▷ Element-wise exponential operation.
9:     $\forall u \in G_1, \forall i \in G_2$
10:     $\mathbf{M}'_{ui} = \frac{\mathbf{M}_{ui}}{\sum_{j=1}^{n_2} \mathbf{M}_{uj}}$                              ▷ Normalize by row.
11:     $\mathbf{M}_{ui} = \frac{\mathbf{M}'_{ui}}{\sum_{v=1}^{n_1} \mathbf{M}'_{vi}}$                              ▷ Normalize by col.
12:     $\beta = \beta * (1 + \beta_r)$                                  ▷ Increase $\beta$.
13: **return** $\mathbf{M}^* \longleftarrow$ greedy_hard_assignment($\mathbf{M}$)

---

*Appendix B.4. GPU Accelerated ARG Matching*

To handle pair-wise matching, we parallelize the process across GPUs to accelerate matching. We implement Line 5–13 of Algorithm A1 with a custom CUDA kernel to process the matching of multiple molecule pairs concurrently. Specifically, each cooperative thread array (CTA) of the GPU is assigned to compute the matching between two molecules. In Algorithm A1, the computation of partial derivative and exponential are element-wise operations is performed. Therefore, we use each thread within the CTA to compute one element in the assignment matrix and all threads work cooperatively to normalize the assignment matrix, which takes advantage of different levels of parallelism on the GPU. We also implement CUDA kernels for computing node- and edge (relation)- similarity and the greedy hard-assignment calculation procedure, so the whole matching algorithm is offloaded onto GPU.

This implementation scales up to a 10-GPU distribution by a workload-partition algorithm, which also alleviates the memory pressure on the GPU. The algorithm follows the principles that no communication between two partitions is needed and the matching of every partition consists of the whole matching result. In this algorithm, we fetch a batch of molecules first, and assign this batch to other non-overlapped batches in the dataset without repeating as different partitions. We perform the matching between molecules from two batches, respectively, in each partition. If there is no unrepeated non-overlapped batches in the dataset, we perform the matching for every molecule in the batch.

*Appendix B.5. Complexity Analysis*

In this section, we analyze the computational complexity of the proposed graph matching method from two aspects: (1) the simplified graduated assignment in Algorithm A1, and (2) the GPU accelerated matching algorithm in Appendix B.4.

The graduated assignment approach for matching ARGs has a low order of computational complexity $O(l_1 l_2)$, where $l_1$ and $l_2$ are the numbers of edges in the graphs. The theoretical computational analysis is discussed in [27,28]. Note that this complexity depends on both the graph size and the sparsity of graphs, that is, the graduated assignment approach becomes more efficient for pairs of sparser graphs. Another worst case analysis of complexity is $O(n_1^2 n_2^2)$, where $n_1$ and $n_2$ represent the numbers of nodes in the graphs. Since $l_1 < n_1^2$ and $l_2 < n_2^2$, the complexity $O(l_1 l_2) << O(n_1^2 n_2^2)$ holds for almost all cases. If

two input graphs are both fully connected, the graduated assignment achieves its worst case of complexity, $O(n_1^2 n_2^2)$. In real scenarios, the graph is usually sparse ($l_1 \propto n_1$ and $l_2 \propto n_2$) and the complexity becomes $O(n_1 n_2)$.

In addition, we take advantage of the massive parallelism of GPU to address the challenge of complexity. The worst-case complexity of graph matching in Algorithm A1 is $O(n_1^2 n_2^2)$. In parallel machines such as GPU, we use parallel-step complexity to asymptotically describe the number of operations performed by threads. In step $s$ of tree reduction, threads perform $\frac{n_1^2 n_2^2}{s^2}$ independent operations. Therefore, the parallel-step complexity is $\mathcal{O}(log(n_1) + log(n_2))$ [62]. Likewise, the matching of all pairs of graphs employs a similar parallelism strategy. In particular, $N$ graphs require $\frac{N(N-1)}{2}$ matching operations, so the parallel-step complexity is $\mathcal{O}(log(N))$. The overall parallel-step complexity for matching $N$ graphs is $\mathcal{O}(log(N)log(n))$, where $n$ is the average number of nodes in the graphs. Therefore, the CUDA-enhanced matching time is sublinear to the number of graphs and graph sizes, which aligns with the results shown in Figure 7.

**Appendix C. Implementation Details and Additional Results**

*Appendix C.1. Settings of ARG Matching*

We used the following settings for the ARG matching Algorithm A1: $\alpha = 0.7$, $\beta_0 = 1$, $\beta_f = 30$, $\beta_r = 0.075$. The node-wise and edge-wise similarity measurements, $s_1(a_u, a_i)$ and $s_2(e_{uv}, e_{ij})$, are task-specific.

In the synthetic data experiment, we defined

$$s_1(a_u, a_i) = \exp(-||a_u - a_i||_2^2)$$

$$s_1(r_{uv}, r_{ij}) = \exp(-3.14 \cdot ||r_{uv} - r_{ij}||_2^2)$$

$$s_1(a_u, a_i) = 1_{ui}$$

$$s_1(r_{uv}, r_{ij}) = 1_{(uv,ij)}$$

On the QM9 dataset, where geometric information, i.e., 3D coordinates for atoms, is equipped, we added bond lengths as edge attributes and the compatibility measurement was designed as

$$s_1(a_u, a_i) = 1_{ui}$$

$$s_1(r_{uv}, r_{ij}) = 1_{(uv,ij)} \cdot \exp(-2||r_{uv} - r_{ij}||_2^2)$$

*Appendix C.2. Training Settings Used in the Synthetic Data Experiment*

The following configurations were applied to all GNN variants. Each baseline model contains two GNN convolutional layers followed by a readout function and then a three-layer MLP to produce predictions. We used a batch size of 32 for the small dataset (500) and 512 for the large one (10,000). We used the cross-entropy loss to train all models and used the Adam optimizer with default initial weights implemented in PyTorch. To prevent overfitting, we used early stopping on the validation loss. We conducted a grid search on the learning rate, batch size and hidden dimension in GNNs. The hyperparameters were tuned as the following: (1) learning rate $\in \{0.1, 0.01, 0.001\}$; (2) hidden dimension $\in \{32, 64\}$; (3) readout function $\in \{max, average\}$; and (4) edge-weight normalization $\in \{True, False\}$.

*Appendix C.3. Additional Results in the Synthetic Data Experiment*

Table A1 shows that MCL-LR is particularly better than GAT, GCN, and GIN at classifying graphs generated from two similar templates, 2 and 5.

**Table A1.** Prediction accuracy for each class on the synthetic dataset.

|  | **Class 1** | **Class 2** | **Class 3** | **Class 4** | **Class 5** |
|---|---|---|---|---|---|
| GAT | $0.710 \pm 0.030$ | $0.495 \pm 0.112$ | $0.950 \pm 0.050$ | $1.000 \pm 0.000$ | $0.515 \pm 0.096$ |
| GCN | $0.920 \pm 0.014$ | $0.766 \pm 0.037$ | $0.857 \pm 0.047$ | $0.897 \pm 0.024$ | $0.686 \pm 0.055$ |
| GIN | $0.886 \pm 0.037$ | $0.296 \pm 0.348$ | $0.955 \pm 0.017$ | $0.940 \pm 0.037$ | $0.668 \pm 0.354$ |
| MCL-LR | $0.996 \pm 0.002$ | $0.996 \pm 0.004$ | $0.994 \pm 0.004$ | $0.998 \pm 0.003$ | $0.999 \pm 0.001$ |

*Appendix C.4. Experimental Settings of Molecular Benchmarks*

The following configurations were applied to all training tasks on the seven molecular benchmarks. We used a batch size of 32 and maximal training epoch of 100. We used the Adam optimizer with a learning rate of 0.001. All experiments were conducted on one Tesla V100 GPU. Before training, we performed data cleaning to remove certain molecules that failed to pass the sanitizing process in the RDKit or contained abnormal valence of a certain atom, as suggested in [63,64]. The detailed dataset statistics are summarized in Table A2. For two motif-level pretraining frameworks, MICRO-Graph [33] and MGSSL [34], they were pretrained on 250k unlabeled molecules sampled from the ZINC15 [65] database and fine tuned on each downstream task. MGSSL performed the same experiments so we tried reproduction based on their available code and optimal model settings. MICRO-Graph did not perform experiments on the datasets we worked on, so we followed the pretraining and finetuning suggestions in MGSSL in reproduction. We were not able to reproduce the same results for MGSSL reported in [34]. Hence, we copy MGSSL's reported results instead of our reproductions.

**Table A2.** Dataset statistics.

| **Dataset** | **# Graphs** | **# Graphs after Cleaning** | **# Tasks** |
|---|---|---|---|
| bace | 1513 | 1513 | 1 |
| bbbp | 2039 | 1953 | 1 |
| clintox | 1478 | 1469 | 2 |
| sider | 1427 | 1295 | 27 |
| tox21 | 7831 | 7774 | 12 |
| toxcast | 8578 | 7245 | 617 |
| hiv | 41,127 | 41,125 | 1 |

*Appendix C.5. Training Settings of MCM+MXMNet on QM9*

To make a fair comparison, we used the same training settings (e.g., training and evaluation data splitting, learning rate initialization/decay/warm-up, exponential moving average of model parameters, etc.) employed in MXMNet [24]. We also kept the same MXMNet configurations (basis functions and hidden layers) as reported in its original paper. The weights of MCM+MXMNet are initialized using the default method in Pytorch. The Adam optimizer was used with the maximal training epoch as 900 for all experiments. The initial learning rate was set to $10^{-3}$ or $10^{-4}$. A linear learning-rate warm-up over 1 epoch was used. The learning rate is then decayed exponentially with a ratio of 0.1 every 600 epochs. To evaluate on valid/test data, the model parameters are the exponential moving average of parameters from historical models with a decay rate of 0.999. Early stopping based on the validation loss was used to prevent overfitting. The motif vocabulary size in MCM was set to 100 or 600. The MCM only adds a small number of parameters (see Table A3).

**Table A3.** Model parameters in DimeNet, MXMNet and MXMNet+MCM.

| Model | # Params |
|---|---|
| DimeNet | 2,100,064 |
| MXMNet | 3,674,758 |
| MXMNet+MCM, vocab_size = 100 | 3,703,302 |
| MXMNet+MCM, vocab_size = 600 | 3,767,302 |

*Appendix C.6. Efficiency of Executing MCM on QM9*

Building motif vocabulary from subgraphs is the most time-consuming part in MCM, especially for large-scale datasets. Hierarchical clustering on a gigantic size of subgraphs is prohibitively expensive. For example, from the QM9 dataset, we obtained 0.5 million one-hop subgraphs. Many of them turned out to be highly similar to each other up to a 3D transformation (rotation + translation). To give an example, the carbonyl functional group (C=O) is quite common in organic compounds. However, the length of the C=O bond in carbonyl may change depending on its local context [66]. To remove "redundancy", we applied a hierarchical clustering technique using average linkage [25], implemented in the Orange3 library [26], to group highly A motifs into the most representative subgraph in a cluster, which has the highest total similarity to the rest of subgraphs in the cluster. For large datasets such as QM9, there is a huge number of subgraphs, which makes the clustering analysis prohibitively expensive for us. To make the computation feasible for QM9, we randomly sampled 40 sets of subgraphs. For each subset, we performed clustering and chose the 1000 most representative subgraphs. Most "redundant" subgraphs were, thus, removed and we obtained a merged subgraph set of size 44,000. We repeated the procedure: divided them into four subsets, performed clustering to obtain 3000 subgraphs for each subset and, finally, obtained a merged set of size 12,000. Another round of single-linkage clustering analysis was applied to the pooled set to find the final 100 or 600 representative subgraphs as the motif vocabulary. We applied the same clustering technique to the 12,000 representative subgraphs in 100 or 600 clusters, and chose one representative subgraph from each cluster as a motif.

To cluster subgraphs using hierarchical clustering, we needed to run a large number of pair-wise matching processes, which took 4.4 h for each subset on 8 RTX 2080 GPUs. Without considering the geometric information such as dataset ogb-molhiv, the graph-matching part is approximately 1.5 times faster. After constructing the motif vocabulary of size 100, then it takes around 13 h to generate the motif matching scores for the whole QM9. Importantly, though, the matching step can be parallelized in a very efficient manner, resulting in significantly lowered computation time. Additionally, the motif vocabulary construction and scoring process only needs to be performed once per dataset. Once constructed, the motif vocabulary can be reused without additional computational expenses.

*Appendix C.7. Additional Results Demonstrating MCM's Interpretability*

Figure A1 shows the 3D visualization of several motifs that represent diverse functional groups, including Fluorophenyl, Trifluoromethyl, Nitrile, Aldehyde, Ester and Methyl. The visualizations confirm that the learned motifs are semantically meaningful and improve the interpretability of our approach.

Table A4 shows the nine local structure categories of the carbons visualized in Figure 6.

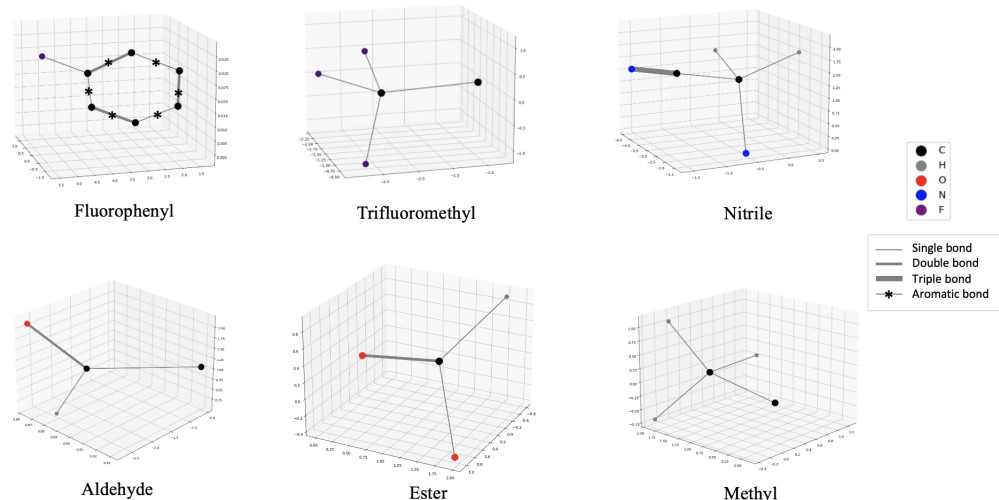

**Figure A1.** 3D visualization of motifs and the functional groups they represent.

**Table A4.** The first column lists the structural abbreviations corresponding to the legends in Figure 6. The second column lists the corresponding chemical groups. The first column shows the structure formula.

| Abbr | Name | Structural Formula |
|---|---|---|
| RPhF | Fluorophenyl | R—⬡—F |
| RCF$_3$ | Trifluoromethyl | F<br>\|<br>R—C—F<br>\|<br>F |
| RCH$_2$OH | Alcohol | H<br>\|<br>R—C—OH<br>\|<br>H |
| RCHO | Aldehyde | O<br>\|\|<br>R—C—H |
| RCOOR′ | Ester | O<br>\|\|<br>R—C—OR′ |
| RCOR′ | Ketone | O<br>\|\|<br>R—C—R′ |
| RCN | Nitrile | R—C≡N |
| RCH$_2$R′ | Methylene | H<br>\|<br>R—C—R′<br>\|<br>H |
| RCH$_3$ | Methyl | H<br>\|<br>R—C—H<br>\|<br>H |

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
