# Peer review of "Motif-Based Graph Representation Learning with Application to Chemical Molecules"

_informatics, doi:10.3390/informatics10010008_

Round 1

Reviewer 1 Report

Dear Authors,

Recently, various GNN techniques have emerged for deep learning on graph-type data. This paper proposed MCM (Motif Convolution Module), a representation layering technique for embedding chemical molecular formulas with graph structures in deep learning, such as GNN. The proposed method calculates the structural similarity through the Motif Convolution Module (MCM) to increase the expressive power of local structural information in the graph. 

At this time, the similarity between ARGs (Attributed Relational Graphs) is calculated to perform clustering for each subgraph effectively. It is claimed that performance can be improved by reflecting semantic information.

However, when calculating the similarity between ARGs to be compared, it seems necessary to explain whether good results can be guaranteed only by calculating how many identical attributes(vocabularies) are included.

In other words, although the paper creates five subgraph templates through Figure 3 and presents experiments on various data sizes through Table 1 as a basis for quantitative evaluation, an experiment on how semantically similar the attributes constituting the subgraphs are seems insufficient.

Of course, in the case of vocabulary used in chemistry, variations such as homonyms and acronyms may be less than those used in general science and technology literature. Still, the semantic similarity between subgraphs from the proposed model was confirmed through experiments using various attributes. It would be best if you showed that you could find it accurately.

Reviewer 2 Report

The use of the Motif Convolution Module (MCM), as a new motif-based graph representation learning technique to better utilize local structural information, is something important to note, although the authors state "In our future work, we will develop motifs that are co -trainable with the rest of a model" would have been remarkable to briefly incorporate some of these models.

The theme is original, relevant, but it does not address a specific gap in the field of ARG, rather it proposes a more efficient alternative to it through MCM.
  The methodology used in the experiment was correct, I applied the two methods (ARG and MCM) to the same specimen and compared the results, I do not consider that additional controls should be added.
  The results and conclusions are solid in terms of the arguments defined in the document and respond to the question posed. As an additional assessment, the experiment could be replicated, but this is outside the review of the document as such.
  I have no comments on the tables and figures, I consider that they are well organized and referenced.
